# THE POTENTIAL OF CoT FOR REASONING: A CLOSER LOOK AT TRACE DYNAMICS

**Gregor Bachmann, Yichen Jiang, Seyed-Mohsen Moosavi-Dezfooli & Moin Nabi**
Apple

## ABSTRACT

Chain-of-thought (CoT) prompting is a de-facto standard technique to elicit reasoning-like responses from large language models (LLMs), allowing them to spell out individual steps before giving a final answer. While the resemblance to human-like reasoning is undeniable, the driving forces underpinning the success of CoT reasoning still remain largely unclear. In this work, we perform an in-depth analysis of CoT traces originating from competition-level mathematics questions, with the aim of better understanding how, and which parts of CoT actually contribute to the final answer. To this end, we introduce the notion of a *potential*, quantifying how much a given part of CoT increases the likelihood of a correct completion. Upon examination of reasoning traces through the lens of the potential, we identify surprising patterns including (1) its often strong non-monotonicity (due to reasoning *tangents*), (2) very sharp but sometimes tough to interpret spikes (reasoning *insights* and *jumps*) as well as (3) at times *lucky guesses*, where the model arrives at the correct answer without providing any relevant justifications before. While some of the behaviours of the potential are readily interpretable and align with human intuition (such as insights and tangents), others remain difficult to understand from a human perspective. To further quantify the reliance of LLMs on reasoning *insights*, we investigate the notion of CoT *transferability*, where we measure the potential of a weaker model under the partial CoT from another, stronger model. Indeed aligning with our previous results, we find that as little as 20% of partial CoT can "unlock" the performance of the weaker model on problems that were previously unsolvable for it, highlighting that a large part of the mechanics underpinning CoT are transferable.

## 1 INTRODUCTION

Chain-of-thought (CoT) reasoning (Wei et al., 2023) has led to several breakthroughs in domains ranging from mathematics (Cobbe et al., 2021; Gao et al., 2023; Luo et al., 2025; DeepSeek-AI et al., 2025) to coding (Chen et al., 2021; Austin et al., 2021; Li et al., 2022; Lozhkov et al., 2024; Rozière et al., 2024), enabling modern language models to now win gold medals at mathematical olympiads (Luong et al., 2025). The underlying idea of CoT is very simple and intuitive: let the model reason through the given problem and explain its steps before giving a final answer. This approach offers two main advantages: (1) Generating additional tokens means more computation available to the model, allowing it to implement more complex routines. (2) CoT enables the model to decompose complex problems into more manageable sub-tasks, akin to human reasoning.

The success of chain-of-thought reasoning is undeniable, yet the precise mechanisms driving it remain poorly understood. A very tempting explanation, due to their (by design) strong resemblance to human reasoning, is that LLMs similarly benefit from spelling out bigger computations more slowly, using techniques such as backtracking and verification to explore several avenues before finally arriving at the best answer (Zhou et al., 2023; Shinn et al., 2023; Madaan et al., 2023b; Press et al., 2023). Other works however suggest that the content of CoTs might not always reflect the actual solving strategy of the model, for instance Lanham et al. (2023); Chen et al. (2025b) show that the model's explanations to addition task do not line up with the underlying computation performed internally. This result seems to rather suggest that CoTs primarily act as computational mechanisms, letting the model execute more complicated algorithms or heuristics "under the hood" while at the same time mimicking human reasoning.

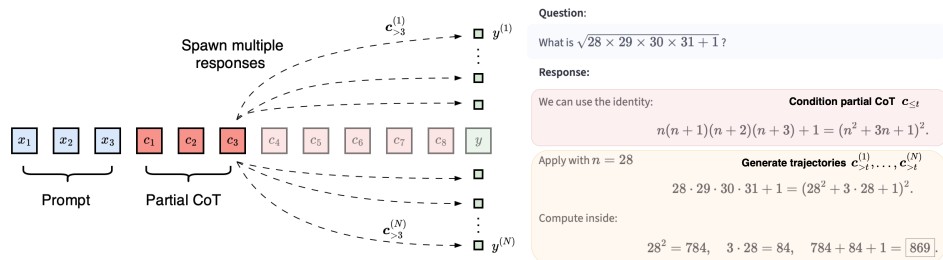

Figure 1: **Left:** Illustration of the calculation of the potential. **Right:** An example prompt and partial CoT, which in this case should intuitively raise the probability of success (i.e. the potential) significantly once discovered or provided by another model.

These perspectives motivate a closer look at how CoT actually contributes in practice. We therefore closely examine reasoning traces produced by several models with a focus on competition-level mathematics questions from AIME-2024, AIME-2025 (MAA, 2025) and MATH-500 (Hendrycks et al., 2021), GPQA-Diamond (Rein et al., 2023) for more general reasoning, as well as coding problems from HumanEval (Chen et al., 2021). In the main text we focus on AIME, as its difficult questions present an ideal arena to study properties of reasoning chains, especially as modern models still produce highly variable CoTs for the same question, *sometimes* leading to the correct solution, but often failing to do so. In this paper, we aim to understand what, or which parts of a CoT make it successful or wrong? To gain insights into this question, we introduce the notion of the *potential*, defined as the probability of success of the model when sampling conditioned on a given partial chain of the CoT (see Eq. 1 for a precise definition). As the potential initially starts out low (models can only sometimes arrive at the right answer), we can use it to monitor precisely which tokens (or collection thereof) increase or decrease it, equipping us with a tool to understand what parts of CoT unlock a previously difficult problem. We observe that similarly to humans, LLMs often exhibit reasoning *insights*, i.e., strong increases of the potential due to the completion of a conceptually difficult step (see e.g., Fig. 1, 5, 9, 11, 12, 13, 14 or 15). Not all spikes in the potential are easily interpretable however; we find that performance can significantly increase through seemingly trivial steps, coined reasoning *jumps* (see e.g. Fig. 5 or Fig. 6) Surprisingly, we observe that the potential is far from monotonic, i.e. not every token contributes effectively towards the final answer but rather long durations of no progress or even sharp drops can occur. The latter are often due to reasoning *tangents*, i.e. approaches which initially look promising but ultimately lead to dead ends or even wrong answers, (see e.g. Fig. 9, 12, 13, 14 or 15).

To further study the usage of reasoning insights in language models, we investigate the degree of *transferability* of CoT between different models. We focus on providing a weaker model with the (partial) CoT from a stronger one, with the motivation that if models indeed struggle with conceptual understanding of the problem, their reasoning might be unblocked when being provided correct sub-steps. Indeed, difficult mathematical questions often involve solving several steps of non-uniform difficulty, with some problems even becoming mostly trivial for humans once a specific insight is obtained or provided. An illustrative example of such a question is shown in Fig. 1, taken from AIME-1989 (MAA, 2025). While the question might look intimidating to many math students at first sight, the problem becomes easily solvable when presented with the insight that $n(n+1)(n+2)(n+3)+1 = (n^2+3n+1)^2$. In other words, human reasoning is often able to transfer if the gap is not too large. For LLMs, we find similar results; problems that were previously unsolved by the weaker model, gradually become solvable as more and more CoT is provided, even as little as 20% of CoT leads to a significant improvement in performance. We observe such transferability even between very different model classes, e.g. Qwen3-0.6B's accuracy significantly improves when provided with partial CoT of GPT-OSS-20B. This suggests that language models can profit from insights provided by stronger models, suggesting that some CoTs work in a model-agnostic way.

## 2  RELATED WORK

Chain-of-Thought reasoning has been very influential in recent years, with every modern language model now being trained to give reasoning-like responses. This characteristic has been strongly

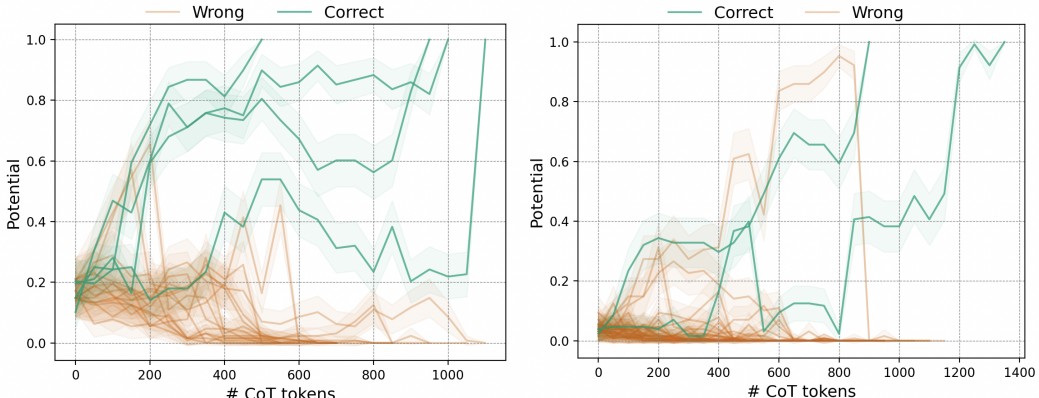

Figure 2: **Potential curves.** Potential of correct and wrong CoTs for Qwen2.5-7B on AIME-2024, Question 5 and 11. Strongly non-monotonic behaviour for both correct and incorrect CoTs.

exacerbated by the emergence of reasoning models such as o1 (OpenAI et al., 2024) and R1 (DeepSeek-AI et al., 2025), further encouraging longer responses by training with reinforcement learning with verifiable rewards. Such models now regularly require generating $128k$ tokens for difficult mathematics questions before returning a final answer. The still human-like nature of these reasoning chains has inspired a surge of works with the aim of interpreting and understanding how these long sequences of tokens actually contribute to the final answer. A line of work has investigated how models react when their CoT is manipulated through insertion of mistakes (Wang et al., 2023) or changes in symbols (Madaan et al., 2023a; Madaan & Yazdanbakhsh, 2022), finding them to be surprisingly robust. Other works have investigated several attribution strategies to identify important parts in CoT (Golovneva et al., 2023; Berchansky et al., 2024; Wu et al., 2023). Opposite types of findings have also been made; Lyu et al. (2023); Lanham et al. (2023); Madsen et al. (2024) have observed that CoT does not always reflect the underlying computation of the model, making it thus difficult to pin-point helpful steps in the first place. Other works go a step further and argue that CoT reasoning should not be compared to human reasoning (Kambhampati et al., 2025; Stechly et al., 2025; Bhambri et al., 2025) or that they outright imitate reasoning without actually performing any (Shojaee et al., 2025). Finally, the line of works most similar to ours also studies conditional generation from partial CoTs; Bigelow et al. (2025) investigate so-called "fork tokens" in the context of neural text generation. Bogdan et al. (2025) also explore the notion of conditional generation to find "thought anchors", parts of CoT that help the model arrive at correct answers. While their focus is on more abstract reasoning concepts such as backtracking and self-verification, we focus on task-relevant insights and also explore the failure modes of CoT reasoning. Finally, Amani et al. (2025) also explore the notion of completing partial CoTs, incorporating the idea in reinforcement learning for better reward signal. Our latter definition of the potential shares strong resemblance to the value function in actor-critique models (Konda & Tsitsiklis, 1999; Sutton & Barto, 2018), similarly measuring the quality of a given state but in a Monte-Carlo fashion.

## 3 POTENTIAL OF COT

**Setup.** Let $\mathcal{V}$ denote the vocabulary. Assume we have a tokenized input prompt $\boldsymbol{x} \in \mathcal{V}^D$ (e.g., encoding a math question) and a ground truth answer $y^* \in \mathcal{V}$ (for simplicity represented by a single token) encoding the expected response (e.g. "513"). Let $\texttt{LM}_\theta$ represent a language model with parameters $\boldsymbol{\theta}$, mapping a sequence of tokens $\boldsymbol{x}$ to the logits of size $|\mathcal{V}|$. When answering to a prompt, models now generate $T \in \mathbb{N}$ intermediate *chain-of-thought* tokens $\boldsymbol{c} \in \mathcal{V}^T$ auto-regressively before arriving at a final answer $y$. I.e. given prompt $\boldsymbol{x}$, we generate $c_t \sim \texttt{LM}_\theta(\bullet|\boldsymbol{c}_{<t}, \boldsymbol{x})$ auto-regressively and only then sample the answer, $y \sim \texttt{LM}_\theta(\bullet|\boldsymbol{c}, \boldsymbol{x})$. Generations involving such intermediate tokens have been observed to outperform models trained (or prompted) to directly provide an answer in a variety of settings (Wei et al., 2023). We will often abuse notation slightly by letting $(y, \boldsymbol{c}_{\geq t}) \sim \texttt{LM}_\theta(\bullet|\boldsymbol{c}_{<t}, \boldsymbol{x})$ denote the (sequential) autoregressive generation, conditional on $(\boldsymbol{c}_{<t}, \boldsymbol{x})$. Typical decoding strategies in language models leverage this stochastic generation and

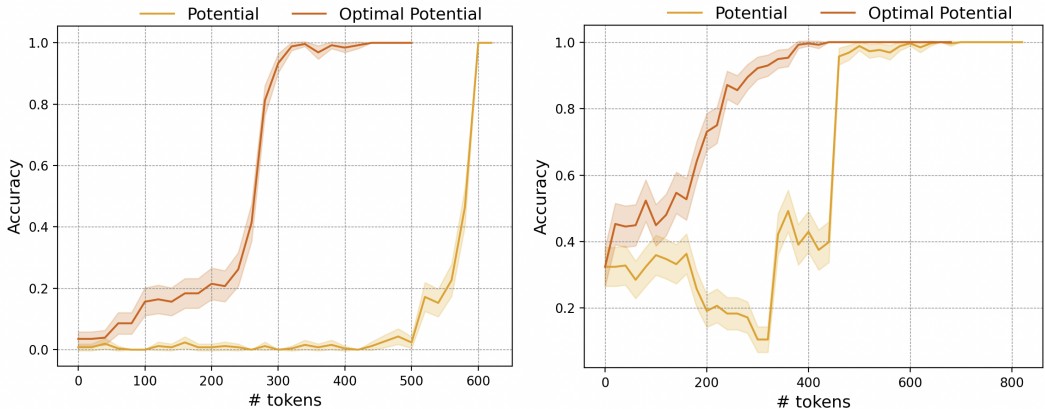

Figure 3: The potential of optimal and standard CoT for AIME-2025-I, question 1 and 5. While standard CoT eventually arrives at the right answer, optimal CoT does so in a more robust way.

it is hence interesting to consider $K \in \mathbb{N}$ such generations by varying the random seeds, either unconditionally or starting from a partial CoT $\boldsymbol{c}_{<t}$,

$$\left(y^{(k)}, \boldsymbol{c}_{\geq t}^{(k)}\right) \overset{i.i.d.}{\sim} \text{LM}_\theta(\bullet|\boldsymbol{c}_{<t}, \boldsymbol{x}) \quad \text{for } k = 1, \ldots, K$$

where we obtain most likely distinct CoT completions $\boldsymbol{c}_{\geq t}^{(k)}$ and final answers $y^{(k)}$.

**Potential.** Given a prompt $\boldsymbol{x}$ and an associated reasoning process $\boldsymbol{c}$ with final answer $y$, it is natural to ask which sub-steps in $\boldsymbol{c}$ contributed most to the overall result. Let us define the *potential* of a chunk of CoT $\boldsymbol{c}_{<t}$ on the prompt $\boldsymbol{x}$ as the probability of correct generation conditioned on $\boldsymbol{c}_{<t}$,

$$\text{pot}(\boldsymbol{c}_{<t}; \boldsymbol{x}) := \mathbb{P}_{(\boldsymbol{c}_{\geq t}, y) \sim \text{LM}_\theta(\bullet|\boldsymbol{c}_{<t}, \boldsymbol{x})} \left(y = y^*\right) \tag{1}$$

Intuitively, if a chunk of CoT is useful or encompasses a step that the model tends to struggle with, conditioning on it should subsequently lead to a higher potential. In mathematical terms, if conditioning on a shorter prefix $\boldsymbol{c}_{<s}$ for $s < t$ has a lower potential compared to $\boldsymbol{c}_{<t}$ i.e. $\text{pot}(\boldsymbol{c}_{<s}; \boldsymbol{x}) < \text{pot}(\boldsymbol{c}_{<t}; \boldsymbol{x})$, this implies that the CoT chunk $\boldsymbol{c}_{s<t}$ "made progress" towards the final solution. On the other hand, if the potential remains similar, $\text{pot}(\boldsymbol{c}_{<s}; \boldsymbol{x}) \approx \text{pot}(\boldsymbol{c}_{<t}; \boldsymbol{x})$, then the chunk of CoT $\boldsymbol{c}_{s<t}$ did not solve a step that is difficult to the model, as it can reliably reproduce it under sampling. This does not necessarily imply that such steps can be skipped as they could entail necessary computations such as a long multiplication, which the model can reliably do but also *needs* to do. Finally, we can have situations where the potential decreases, with CoTs actively worsening the state of the model. On average however, we can show mathematically that the potential improves monotonically over all correct CoTs:

**Proposition 1.** *Conditional on the event that the full CoT $\boldsymbol{c}_{1:T}$ yields the correct final answer $y^*$, it holds for every $t \leq T$ that*

$$\mathbb{E}\left[\text{pot}(\boldsymbol{c}_{<t}; \boldsymbol{x})\right] \leq \mathbb{E}\left[\text{pot}(\boldsymbol{c}_{<t+1}; \boldsymbol{x})\right].$$

We invite the reader to check the proof in Appendix A.2. Hence on average, every token $c_t$ should push the potential higher, encouraging the model to converge towards the correct solution, reflecting the intuition that chain-of-thought performs *evidence accumulation*. Calculating the potential exactly is unfortunately intractable, so in practice we use the following estimator instead,

$$\text{pot}_N(\boldsymbol{c}_{<t}; \boldsymbol{x}) := \frac{1}{N} \sum_{n=1}^{N} \mathbb{1}_{\{y^{(n)} = y^*\}} \quad \text{where } \left(y^{(n)}, \boldsymbol{c}_{\geq t}^{(n)}\right) \sim \text{LM}_\theta(\bullet|\boldsymbol{c}_{<t}, \boldsymbol{x})$$

Sampling a higher number of trajectories $N$ will provide a better approximation to the true potential. We observe that setting $N = 128$ gives very reliable estimations of the potential and use it throughout this work. We provide more experimental details regarding the computational complexity in Appendix A.5.

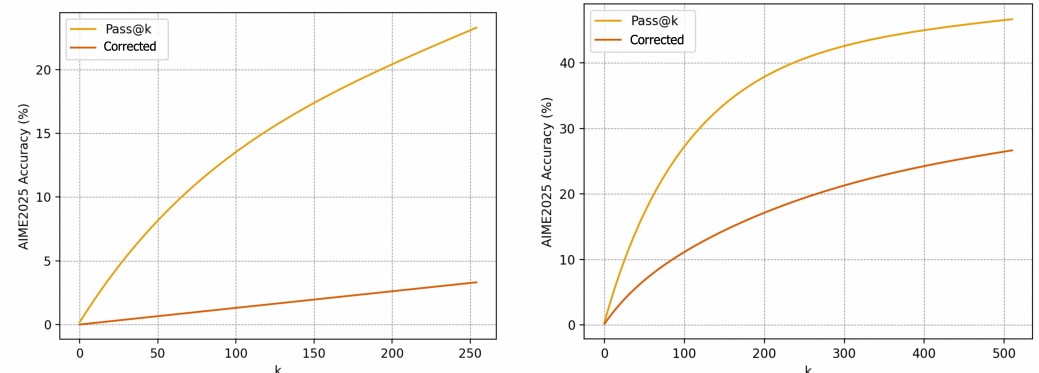

Figure 4: **Inflated** $\text{pass@}k$. We show $\text{pass@}k$ accuracies and the corresponding corrected values for Qwen2.5-1.5B (left) and Qwen2.5-7B (right).

## 4 SHAPE OF POTENTIAL CURVES

We now empirically study the potential $\text{pot}(\boldsymbol{c}_{<t}; \boldsymbol{x})$ as a function of the CoT chunk length $t$. When conditioning on CoTs $\boldsymbol{c}$ that lead to the correct answer $y = y^*$, based on Prop. 1, we expect the potential to be a smooth and monotonic function in $t$, with every chunk of CoT $\boldsymbol{c}_{s<t}$ positively contributing to the overall solution. We will mainly focus on difficult competition-level mathematics questions, where the potential $\text{pot}(\boldsymbol{c}_{<0}; \boldsymbol{x})$ corresponding to the "empty" CoT $\boldsymbol{c}_{<0}$ is strictly between 0 and 1, i.e. the model only sometimes produces the correct answer when prompted from scratch. If the model is always correct, the potential does not offer any insight into the CoT; all steps are equally easy to the model. In contrast, if performance starts significantly lower, we can precisely pinpoint where a successful CoT overcame hurdles that stopped most other attempts. We calculate the potential curves for a variety of models, including both the non-thinking types of models Qwen2.5 (sizes 1.5B and 7B), (Qwen et al., 2025) and Llama-3.1 (sizes 8B and 70B) (Grattafiori et al., 2024), as well as the reasoning models Qwen3 (sizes 0.6B and 32B) (Yang et al., 2025). We display a variety of potential curves (both for correct and wrong trajectories) in Fig. 2 for two samples taken from AIME-2024. Surprisingly, typical chain-of-thought exhibits quite erratic potentials, with certain sections of CoT actively worsening the probability of success, going against the theoretical result in Prop. 1. We will examine the characteristics of potential curves qualitatively in close detail in Sec. 5. We quantify the following properties of potential curves often exhibited across AIME-2024: (1) Very sharp increases in the potential in a small token window, we will later refer to these occurrences as reasoning *insights* and *jumps*. (2) Very sharp drops in the potential, we coin this behaviour reasoning *tangents* or *flaws*. (3) Extremely late increases in the potential, which previously remained flat and close 0. We will show qualitatively in Sec. 5 that such CoTs are very often associated with *guessing*, i.e. the model produces a correct answer without relying on its previously generated reasoning and at times even admits to do so.

| MODEL | REASONING | INSIGHTS ↑ | TANGENTS ↓ | LATE SPIKE | MONOTONICITY |
|---|---|---|---|---|---|
| QWEN2.5-1.5B | ✗ | 40% | 5% | 20% | 45% |
| QWEN2.5-7B | ✗ | 62% | 9.5% | 14% | 42% |
| LLAMA3.1-8B | ✗ | 46% | 33% | 6% | 15% |
| LLAMA3.1-70B | ✗ | 37% | 40% | 5% | 17% |
| QWEN3-0.6B | ✓ | 55% | 41% | 10% | 15% |
| QWEN3-32B | ✓ | 36% | 18% | 0% | 36% |

Table 1: Behaviours of potential for several reasoning and non-reasoning models on AIME-2024.

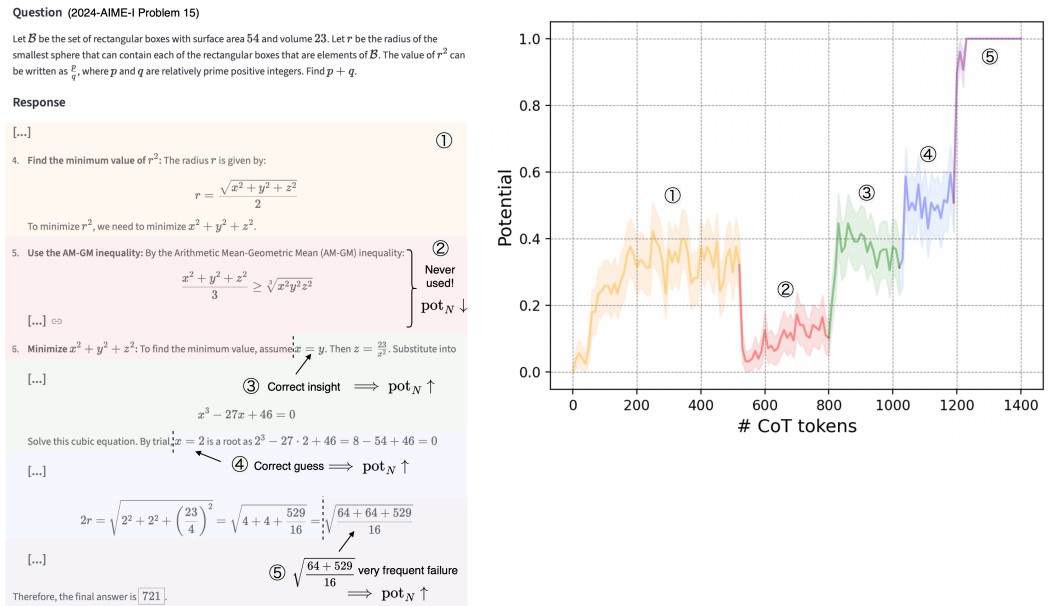

Figure 5: **Reasoning tangents and insights.** Qwen2.5-7B's potential $\text{pot}_{256}(\bullet; \boldsymbol{x})$ behaving strongly non-monotonic. The reasoning *tangent* ② hurts the potential, while the reasoning *insights* ③ (observing the symmetry $x = y$ of the problem) and ④ (finding the root of the cubic equation) push the potential back on track. Finally, the model performs a reasoning *jump* ⑤ (for some non-obvious reason, this particular calculation is difficult for the model).

**Quantifying the shape.** In the following, we will derive some quantitative summary statistics corresponding to the observations we made based on the plots in Fig. 2. We calculate the potentials for 128 responses per sample on AIME-2024 (total of $30 \times 128$ samples) and filter out responses that led to wrong answers. We further only consider samples that are difficult enough for the given model to not reach perfect accuracy without any partial CoT. We then derive four summary statistics that aim to describe the properties introduced above, we detail their precise definitions in Appendix A.7. We show analogous results on MATH-500 in Table 2 in the Appendix.

We display the results in Table 1. Our initial observations are substantiated; only half of the CoTs exhibit monotonicity, with reasoning models tending to produce even more erratic potentials. Non-reasoning models seem to exhibit more late spikes, which aligns with our qualitative observations later in Sec. 5 that such models tend to produce correct answers often through guessing on very difficult problems. Model size also seems to surpress this behaviour more, which is expected since larger models generally tend to perform better. Reasoning tangents occur more often for reasoning models, aligning well with the observation in the literature that such models have the tendency to *overthink* (Chen et al., 2025a), i.e. they discard the discovered, correct answer and explore alternative but flawed approaches. This also partially explains their less monotonic potential. All models exhibit a high amount of reasoning insights, suggesting that most of the difficulty is concentrated in a few key steps instead of being uniformly spread out, more akin to human reasoning.

**Amount of guessing.** We now focus on the situations where the potential spikes very late in the reasoning process, which we connect almost uniquely with the model *guessing* answers correctly. This has a strong effect on the $\text{pass}@k$ metric used to assess model capabilities, which we show is strongly impacted in the case of Qwen2.5-1.5B and 7B when used in conjunction with weak verification such as final answer verification in mathematical benchmarks. For a dataset consisting of $P$ queries $\{\boldsymbol{x}_i\}_{i=1}^{P}$ with corresponding answers $\{y_i^*\}_{i=1}^{P}$, we sample $k$ responses $y_i^{(j)}$ per question from the model and measure if the correct answer is at least once among this set, i.e.

$$\text{pass}@k = \frac{1}{P} \sum_{i=1}^{P} \mathbb{1}_{\left\{y^* \in \{y_i^{(1)}, \ldots, y_i^{(k)}\}\right\}}$$

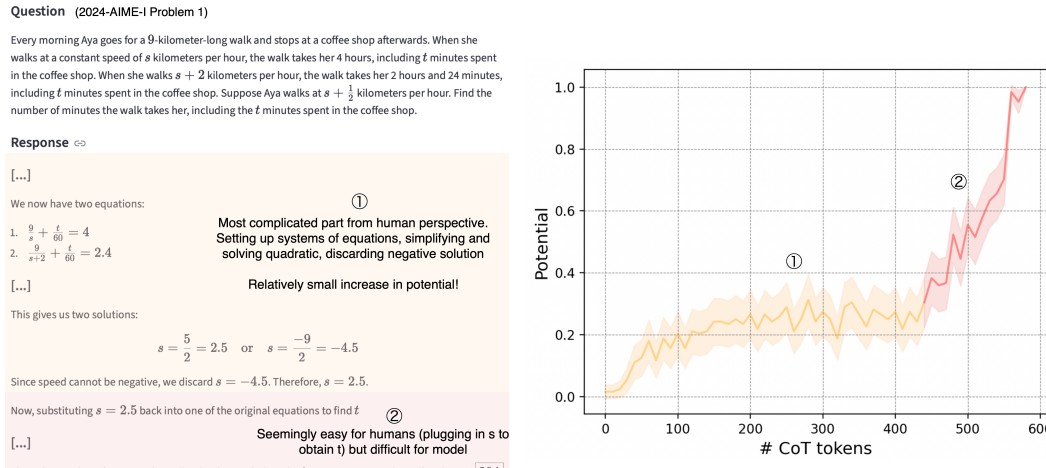

Figure 6: **Unaligned difficulty.** Qwen2.5-1.5B solves most difficult parts in ① but only small increase in potential. Seemingly easier part ② of just obtaining $t$ given $s$ and adding the two turns out to be significantly more difficult.

The idea behind this metric is to measure whether the model does possess the ability to sometimes achieve the right answer, albeit not reliably. Especially for large $k$, this metric could fall victim to lucky guesses as (1) it only takes one correct answer to obtain the full score and (2) the reasoning process is usually not being assessed in the case of mathematics benchmarks. Indeed, in Fig. 4 we show that the $\text{pass@}k$ scores can be very inflated by flagging samples with the *late spike* statistic, in this case on AIME-2025.

**Optimizing the potential.** Given our observation that CoT does not naturally follow a monotonic curve, with many tokens even worsening performance, the following question emerges:

*Can we search the space of CoT $\boldsymbol{c}$ such that every sub-CoT $\boldsymbol{c}_{s<t}$ contributes?*

One way to try and maximize the potential of every chunk of CoT is to set a chunk size $C \in \mathbb{N}$ and randomly explore candidate chunks, calculate their potential and keep the highest scoring chunk. In this manner, we can construct a CoT that increases the potential at least gradually if the model admits such reasoning, ideally avoiding issues such as reasoning tangents. We summarize the recipe in Algorithm 1 and the incurred computational complexity in Appendix A.6 more formally. We indeed find that models admit such optimized CoT, we display some associated potential curves in contrast with regular CoT in Fig. 3. We can indeed see that the optimized CoT displays strong monotonicity with most tokens contributing to the potential. This is in stark contrast with the standard CoT, which either does not increase the potential for a long token horizon (left side of the figure), or even actively worsens it (right side). We examine such CoTs more qualitatively in Appendix A.3.

## 5 A CLOSER LOOK AT CHAIN-OF-THOUGHT REASONING

We now perform a qualitative analysis of various chain-of-thought reasonings on competition-level mathematics. Due to the verbosity of reasoning models such as Qwen3, we limit this section to the Qwen2.5 series, whose CoT is more concise and thus more amenable to direct interpretation. The only exception is Fig. 7, where we display parts of a trace from Qwen3-0.6B. Our goal is to precisely align the potential curve with the underlying reasoning produced by the model, and as a consequence obtain an understanding of the types of tokens that drive or hinder the progress. For space reasons we defer from displaying the full CoT but instead show only the sections crucial to the potential. We refer the interested reader to Appendix A.3 for additional qualitative examples.

We display the first sample obtained from Qwen2.5-7B in Fig. 5, a potential curve that exhibits strong non-monotonicity as we have often encountered (see Fig. 2). We dissect the reasoning into five segments according to the potential. Segment ① steadily makes progress towards the solution

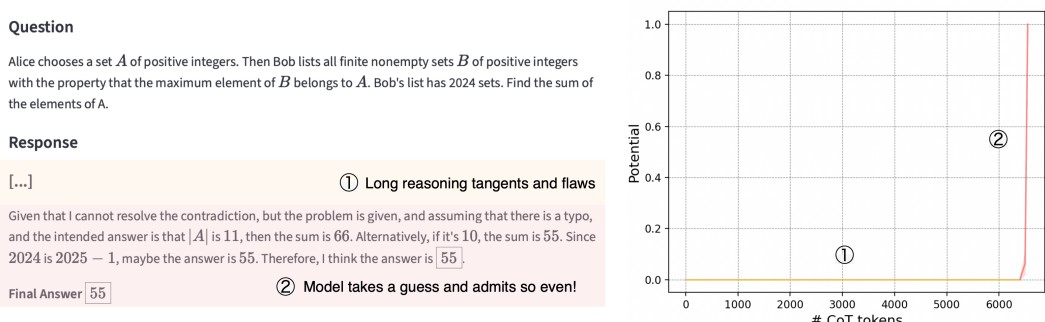

Figure 7: **Reasoning tangents and guessing.** Qwen3-0.6B goes on a long reasoning tangent in ①
that does not increase the potential over a long token horizon. Finally it outputs a final answer in ②,
itself admitting that the guess is not backed by the reasoning prior but seems likely to the model.

by correctly expressing the radius as a function of the sides of the box and formulating the optimiza-
tion problem. In segment ② the model goes on a reasoning *tangent*, a step that is not necessarily
wrong but happens to not work out for the particular problem (*AM-GM inequality* gives a non-tight
lower bound for the minimum). The model manages to ignore this step in this particular trajectory,
but on average suffers from this distraction, leading to a sharp drop in the potential. In segment ③
and ④, the model correctly recognizes the symmetry of the problem as well as discovers the root of
the cubic equation, with both *insights* consequently boosting the accuracy akin to human reasoning.
Finally, in segment ⑤ we observe the final spike in the potential, stemming from a simple arith-
metics step that the model tends to get wrong. While the previous spikes were readily interpretable,
the last one seems more unintuitive, given that the model manages to very reliably perform the ar-
guably harder arithmetics steps just before. We coin this a reasoning *jump*, a very sharp increase in
the potential that largely seems due to a very model-specific issue.

Such misalignment in perceived difficulty of sub-steps is often present in CoT, in Fig. 6 we dis-
play another reasoning trace of Qwen2.5-7B along with the associated potential which exhibits this
surprising characteristic. Segment ① here does the conceptual heavy-lifting; it correctly deduces
the associated system of equations in two variables, simplifies and obtains the solution for the first
variable $s$. The completion of these seemingly involved steps is only rewarded with a small increase
in potential, as opposed to humans, the model does not struggle here. Instead, the more difficult
steps contained in segment ② consist of now obtaining the second variable $t$, which only involves
plugging the value for $s$ into the previously derived equation. Compared to the previous segment,
finishing the problem starting from the end of ① would be a significantly simpler task for humans.

Another surprising insight we obtained is that models can be very capable of guessing solutions to
such problems. In Fig. 7 (and Fig. 10) we display the reasoning of Qwen3-0.6B. While the content
of segment ① at first sight looks relevant, closer inspection reveals that the final answer "80" is not
at all deduced from the reasoning performed before. The answer seems to be a lucky guess, most
likely informed by the fact that answers to such competition-level questions usually take the form of
an integer value. This guessing is elegantly reflected in the potential curve; the reasoning in segment
① (which essentially encompasses the entire CoT) does not make any progress at all towards the
final answer, precisely because the model is most likely making a guess in the end, which more often
than not ends up being wrong.

## 6 Transferability of CoT Through the Lens of Potential

Motivated by the insights from Sec. 4 and 5, we now investigate if reasoning *insights* transfer be-
tween different families of models, which would further underscore that the mechanisms underlying
CoT reasoning share parallels with human reasoning. We hypothesize that if the sub-steps present
information gain through reasoning insights, (similar to e.g. Fig. 1), weaker models could be able to
solve problems that were previously too difficult. We study this scenario for both reasoning and non-
reasoning models. In the first setup we consider Qwen3-0.6B as the weak model, which is provided
with partial CoT from its bigger version Qwen3-32B. We also explore traces from GPT-OSS-20B to

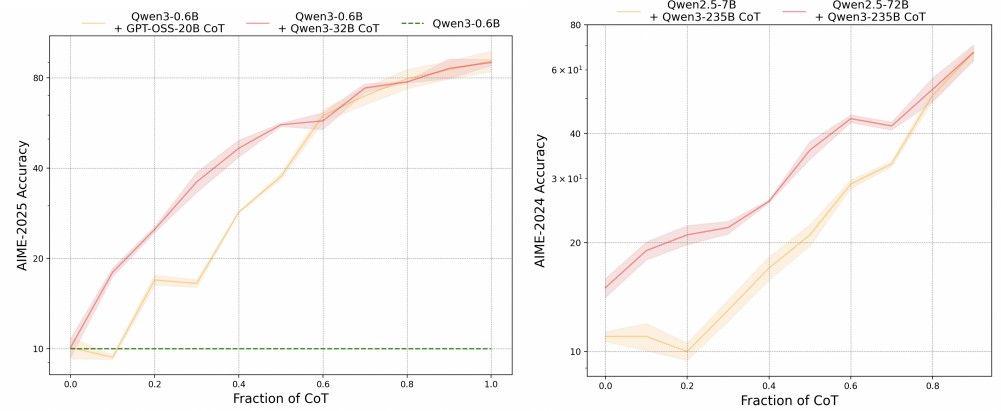

Figure 8: **Transferability of CoT. Left:** Accuracy on AIME-2025 of weaker reasoning model Qwen3-0.6B when provided with partial CoT from Qwen3-32B (red) and from GPT-OSS-20B (orange), leading to very quick improvements. **Right:** Accuracy of non-reasoning models Qwen2.5-7B and Qwen2.5-72B when provided with a partial CoT based on the final summary output of reasoning model Qwen3-235B.

further assess how robust transferability is with respect to out-of-distribution scenarios. For the non-reasoning models we instead create a dataset of *gold* CoT, using one of the strongest public models Qwen3-235B to produce answers on AIME-2024 in thinking mode. We then extract the CoT after thinking, which presents a clean summary of the long thinking traces and use these as partial traces. We then test the weaker Qwen2.5-7B and Qwen2.5-72B models, letting them complete the partial responses for various percentages. We display the resulting test accuracies as a function of the fraction of partial CoT in Fig. 8. We observe that surprisingly, in both reasoning and non-reasoning scenarios, the models manage to not only maintain their original accuracies but quickly improve (with as little as 20% CoT), answering previously unsolved questions. While the CoT does seem to transfer better within the same family, Qwen3-0.6B can still leverage the significantly different traces from GPT-OSS-20B, suggesting that the mechanisms driving the performance are universally shared between models to a strong degree.

## 7 CONCLUSION

In this work we have investigated chain-of-thought reasoning in large language models through the notion of the associated potential. We have performed an in-depth analysis of parts of CoT that strongly move the potential upwards (reasoning insights and jumps), as well as tokens that actively worsen the performance due to reasoning tangents. We further investigate reasoning insights by introducing the notion of CoT transferability, which measures to what degree a weaker model can profit from the partial CoT of a stronger one. We show that the insights of the stronger model indeed help push the performance of the weaker one beyond what it can typically solve on its own, highlighting that CoT indeed relies on such interpretable mechanisms.

We believe that the potential can be used in future work to further understand how CoT contributes to successful completions of LLMs, helping to pin-point its important parts within the very long chains of today's reasoning models. The potential could also serve as a partial reward in RL training to perform more fine-grained credit assignment, indeed concurrent work has already started exploring this angle (Guo et al., 2025; Hou et al., 2025). Finally, we also hope that the transferability of partial CoTs can be leveraged in reinforcement learning to reduce sparsity of rewards with Amani et al. (2025) already obtaining positive initial results.

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

## A  APPENDIX

### A.1  EXPERIMENTAL DETAILS

We use vllm (Kwon et al., 2023) for all of our experiments. For potential calculation we set $N = 128$ and use a temperature of $T = 0.6$ and $p = 0.95$ as sampling parameters. For all models and datasets we generate $T = 32k$ tokens excluding the prompt. To ensure that the potential does not increase due to higher generation length, we always subtract the length of the partial CoT from $32k$ and use this number as $T$.

### A.2  PROOF OF PROPOSITION 1

Here we present the previously omitted proof of Proposition

By Bayes' rule, for any token $c_{t+1}$ we have

$$f_{t+1} = \mathbb{P}(y = 1 \mid x, c_{1:t}, c_{t+1}) = \frac{f_t\, p_1(c_{t+1})}{f_t\, p_1(c_{t+1}) + (1 - f_t)\, p_0(c_{t+1})}.$$

Taking expectation with respect to $c_{t+1}$ drawn from $p_1$, i.e. conditioned on the event that the rest of the run is correct, gives

$$\mathbb{E}[f_{t+1}] = f_t \sum_{c_{t+1}} p_1(c_{t+1})\, \frac{p_1(c_{t+1})}{f_t p_1(c_{t+1}) + (1 - f_t)p_0(c_{t+1})}.$$

Equivalently,

$$\mathbb{E}[f_{t+1}] = f_t \sum_{c_{t+1}} \frac{p_1(c_{t+1})^2}{f_t p_1(c_{t+1}) + (1-f_t)p_0(c_{t+1})}.$$

Now apply the Cauchy–Schwarz inequality with weights $q(c_{t+1}) = f_t p_1(c_{t+1}) + (1-f_t)p_0(c_{t+1})$:

$$\left(\sum_{c_{t+1}} \frac{p_1(c_{t+1})^2}{q(c_{t+1})}\right)\left(\sum_{c_{t+1}} q(c_{t+1})\right) \geq \left(\sum_{c_{t+1}} p_1(c_{t+1})\right)^2.$$

Since $\sum_{c_{t+1}} q(c_{t+1}) = 1$ and $\sum_{c_{t+1}} p_1(c_{t+1}) = 1$, it follows that

$$\sum_{c_{t+1}} \frac{p_1(c_{t+1})^2}{q(c_{t+1})} \geq 1.$$

Therefore,

$$\mathbb{E}[f_{t+1}] \geq f_t.$$

Finally, taking expectation over prefixes $c_{1:t}$ distributed as on correct runs yields

$$\mathbb{E}[f_{t+1}] \geq \mathbb{E}[f_t],$$

which is the desired result.

### A.3 MORE COT EXAMPLES

We display more quantitative results in Table 2 for the MATH-500 dataset, in Table 3 we show results for the coding benchmark HumanEval (Chen et al., 2021) and in Table 4 the ones for GPQA-Diamond (Rein et al., 2023). We find that the same trends observed for AIME-2024 hold also in this case. Interestingly, models seem more stable on coding benchmarks, with their CoT displaying more monotonic behaviour and in general less tangents. For GPQA we find trends more consistent with AIME, most likely because more questions are very difficult compared to HumanEval. As expected due to GPQA being a multiple choice benchmark with only four choices, we observe a significantly higher guessing rate. In this case we adjusted the threshold for guessing to $25\%$ (as opposed to the previously used $5\%$) as a random guessing baseline would of course always achieve $25\%$.

We also show more annotated CoT in Fig. 7, Fig. 9 and for coding in Fig. 14 and Fig. 15. In Fig. 9 we have again have the model performing a reasoning insight, correctly realizing that the exponents can be deduced from the binary representation of the number. We then finally have a reasoning jump, where the model experiences a strong boost in potential from the word "correspond". While at first sight not clearly interpretable, we hypothesize that this word forces the model to output concrete values for $a_i$'s, otherwise a common failure model as the model tries to further refine their computation.

| MODEL | REASONING | INSIGHTS ↑ | TANGENTS ↓ | LATE SPIKE | MONOTONICITY |
|---|---|---|---|---|---|
| QWEN2.5-1.5B | ✗ | 23% | 11% | 1% | 40% |
| QWEN2.5-7B | ✗ | 24% | 12% | 1% | 41% |
| LLAMA3.1-8B | ✗ | 19% | 21% | 1.8% | 31% |
| LLAMA3.1-70B | ✗ | 22% | 23% | 1.5% | 30% |
| QWEN3-0.6B | ✓ | 30% | 40% | 1.4% | 30% |
| QWEN3-32B | ✓ | 28% | 35% | 0.9% | 45% |

Table 2: Behaviours of potential for several reasoning and non-reasoning models on MATH-500.

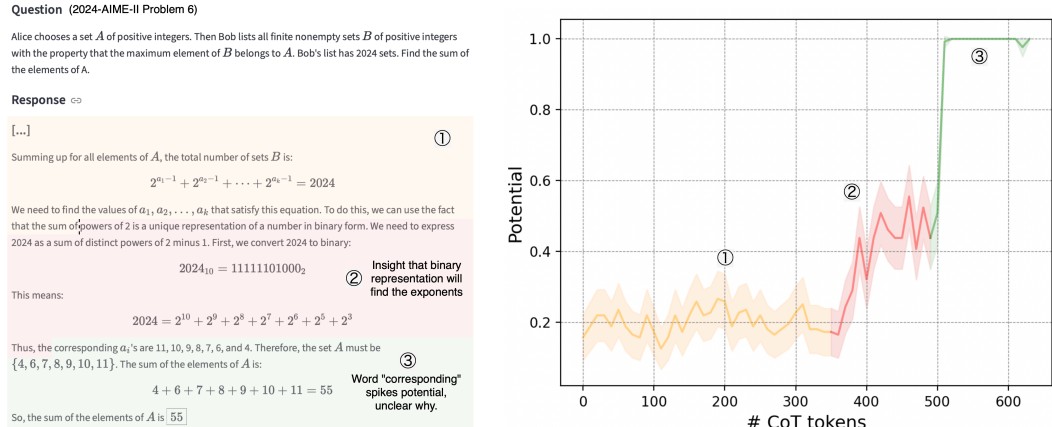

Figure 9: **Unintuitive reasoning jumps.** Qwen2.5-7B's potential $\text{pot}_{256}(\bullet; \boldsymbol{x})$ remains flat in ① although crucial insights are obtained. The potential then increases due to a reasoning *insight* in ② (realizing that the binary representation determines the exponents). In ③ we obtain the final spike at the word "corresponding", a reasoning *jump*, which seems strange from a human perspective. We hypothesize that it might force the model to output values for $a_i$'s, which indeed is the next logical step. We indeed observe that without this word, the model continues to perform unnecessary calculations, subsequently leading to wrong values for $a_i$.

| MODEL | REASONING | INSIGHTS ↑ | TANGENTS ↓ | LATE SPIKE | MONOTONICITY |
|---|---|---|---|---|---|
| QWEN2.5-1.5B | ✗ | 39% | 7% | 0% | 55% |
| QWEN2.5-7B | ✗ | 30% | 4.8% | 0% | 73% |
| LLAMA-3.1-8B | ✗ | 41% | 16.5% | 0.4% | 42.1% |

Table 3: Behaviours of potential for several reasoning and non-reasoning models on the coding benchmark HumanEval.

| MODEL | REASONING | INSIGHTS ↑ | TANGENTS ↓ | LATE SPIKE | MONOTONICITY |
|---|---|---|---|---|---|
| QWEN2.5-1.5B | ✗ | 17% | 6.8% | 19% | 20.9% |
| LLAMA-3.1-8B | ✗ | 29.1% | 20.0% | 17.6% | 25.8% |

Table 4: Behaviours of potential for several reasoning and non-reasoning models on the coding benchmark GPQA-Diamond.

In Fig. 10 we again observe a reasoning guess from Qwen2.5-1.5B, where the CoT in segment ①, while seemingly making sense at first sight, actually does not contribute to the final answer at all. In fact the number 80 does not relate at all to the computations made before. This is reflected in the potential, that shows a spike only towards the very end, highlighting that the CoT indeed did not contribute.

Finally, we show an instance of optimized CoT introduced in Sec.4. We observe that the potential is now strongly monotonic, with almost every partial CoT leading to some improvement in the potential. This is also reflected qualitatively, we can see that the CoT is more concise in language, in fact we can display all of it here. In segment ① the model makes slower progress as those are steps it can reliably do. Finally, the model undergoes a reasoning insight ② with the model discovering that $d$ needs to divide 56.

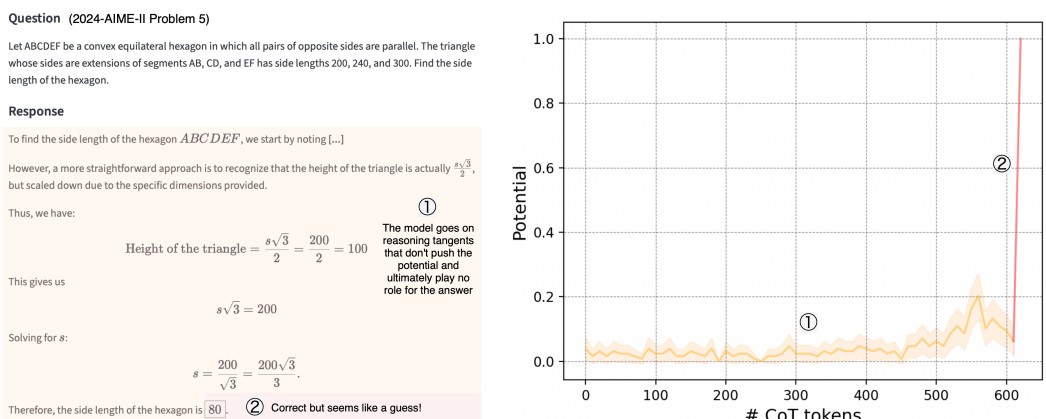

Figure 10: **Reasoning tangents and guessing.** Qwen2.5-1.5B goes on a long reasoning tangent in ① that does not increase the potential over a long token horizon. Finally it outputs a final answer in ② unrelated to the previous reasoning that happens to be correct.

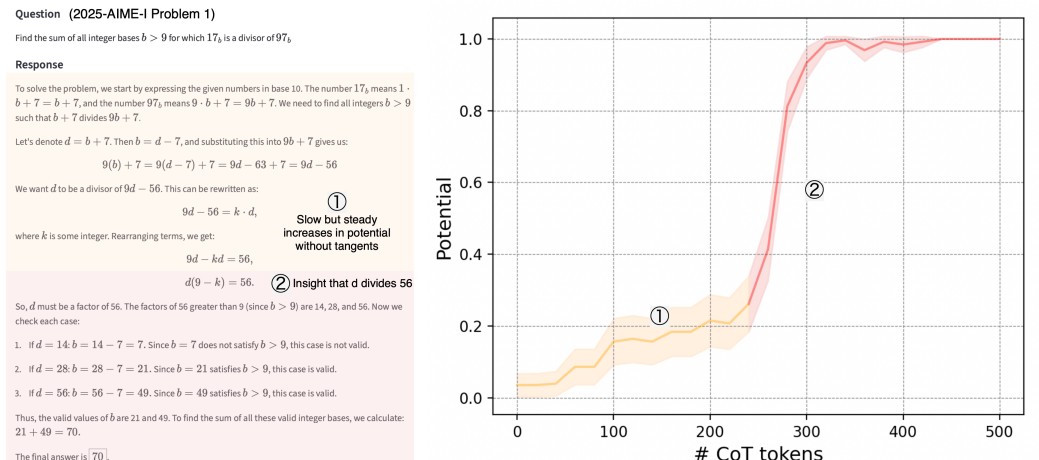

Figure 11: **Optimized CoT.** We show a trajectory based on the optimized CoT from Qwen2.5-1.5B. The CoT is more concise, actually allowing us to show it here in full length. The potential is monotonic as anticipated and all tokens contribute to it. In segment ① the model makes slower progress as those are steps it can reliably do. Finally, the model undergoes a reasoning insight ② with the model discovering that $d$ needs to divide $56$.

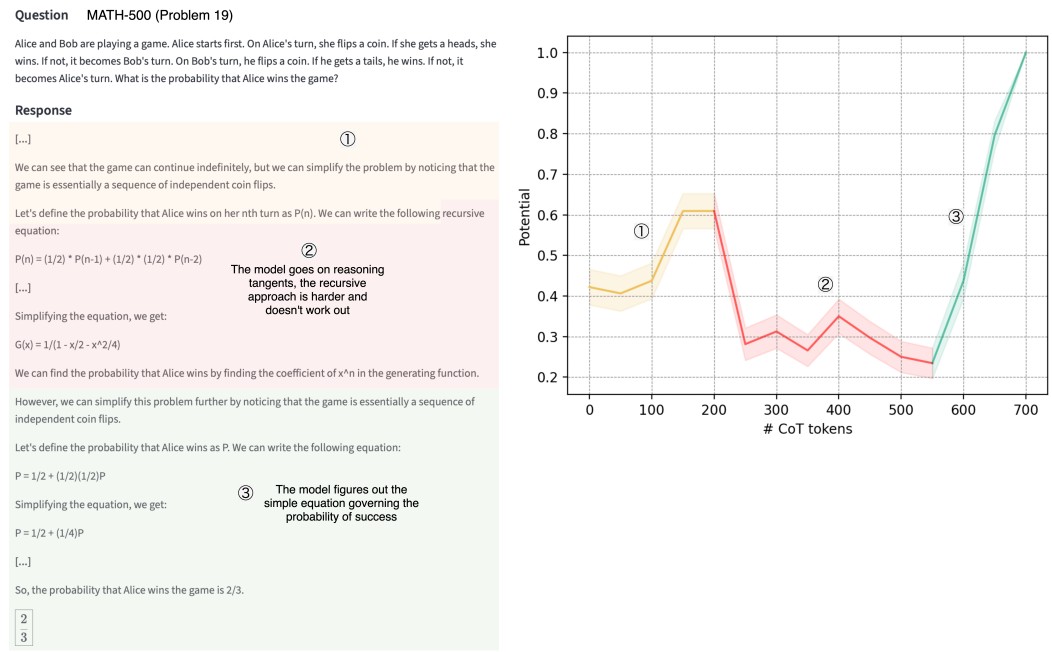

Figure 12: **Reasoning tangents and insights for Llama3.1-8B on Math-500.** We show an example for another dataset using Llama3.1-8B. Similar observations hold; the models tend to go on reasoning tangents as shown here with the model exploring a more difficult recursive approach, which it ultimately abandons by deriving the key insight, expressing the probability as a simple linear equation.

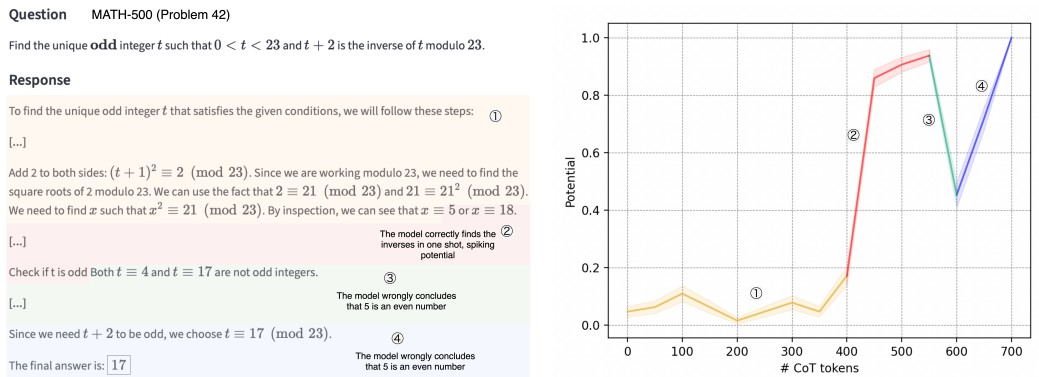

Figure 13: **Reasoning tangents and insights for Llama3.1-8B on Math-500.** We show an example for another dataset using Llama3.1-8B. Again the model displays a key insight, realizing directly that $5$ and $18$ are the square roots. A small tangent is also encountered, with the model missing that $17$ is an odd number, but subsequently corrected later in the reasoning.

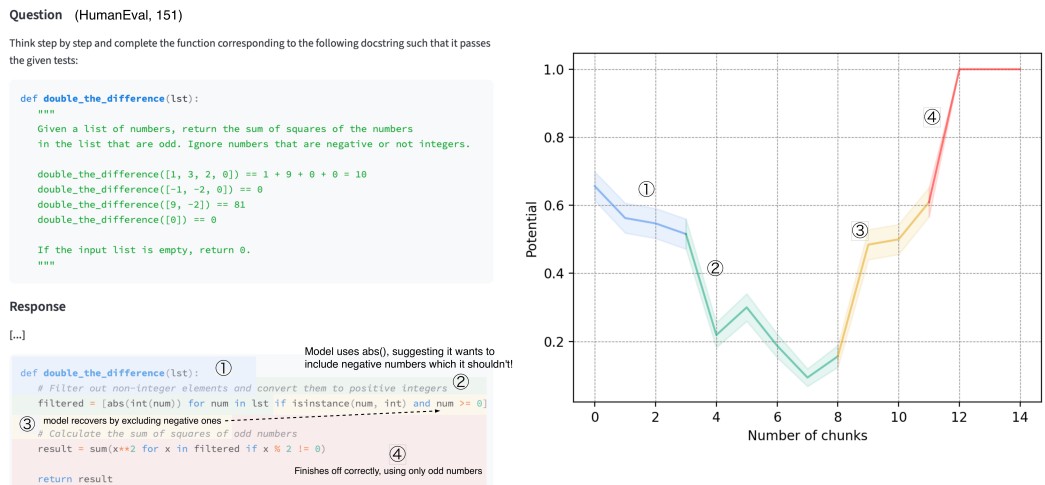

Figure 14: **Potential for coding tasks.** We analyze a trace of Qwen2.5-1.5-Instruct on HumanEval and identify the occurence of tangents and insights. Here, the model gets distracted initially by outlining that it will convert all numbers to positive integers, and indeed using commands abs() and int(), dropping the potential significantly as a consequence. The model then recovers by fixing the for-loop using if statements, which actually render abs() and int() effectless as the loop anyways only goes over positive integers.

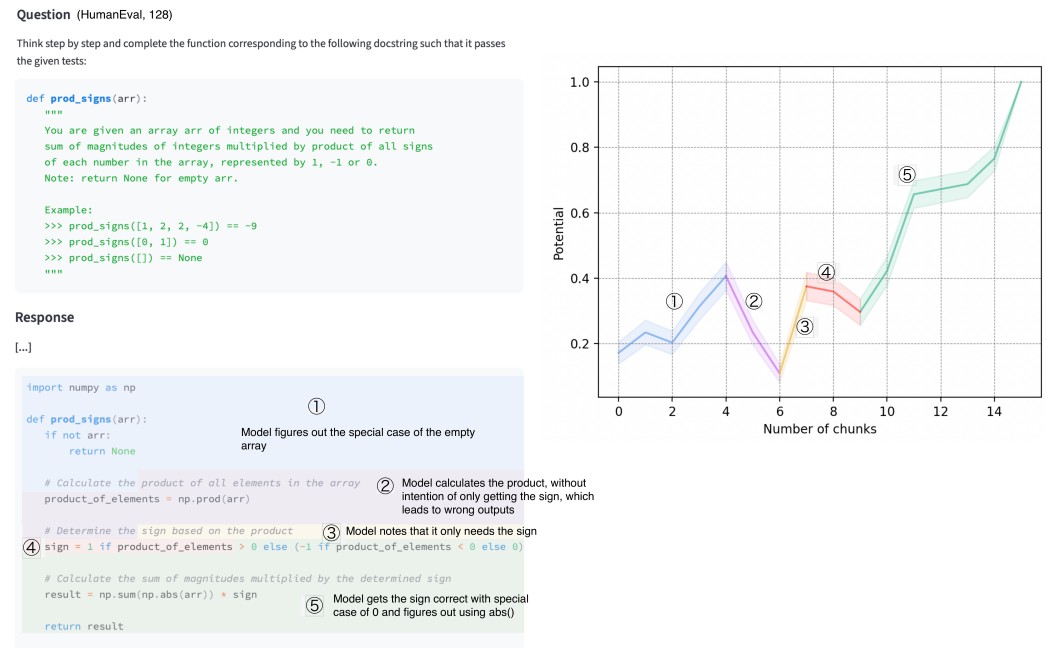

Figure 15: **Potential for coding tasks.** We analyze a trace of Qwen2.5-1.5-Instruct on HumanEval and identify the occurence of tangents and insights.

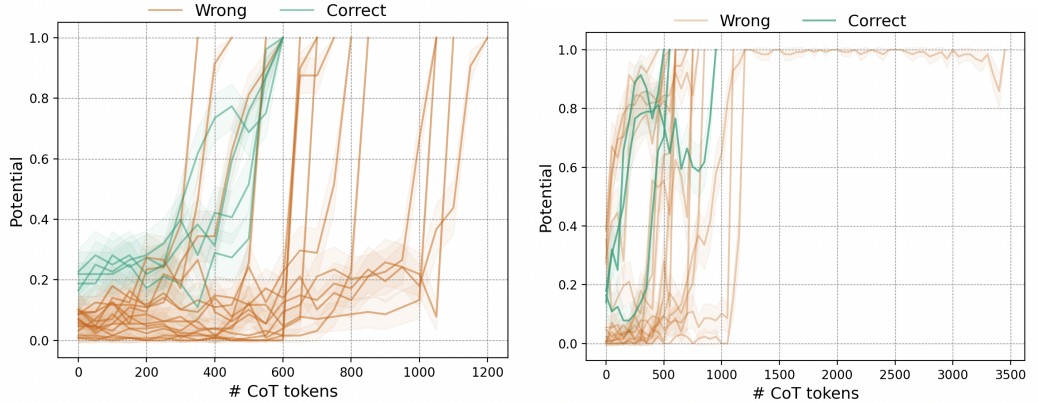

Figure 16: **Stability profiles.** Stability profiles for Qwen2.5-1.5B and Qwen2.5-7B on AIME 7 and 26 respectively. Correct and wrong answers exhibit similar profiles across models and questions.

## A.4   STABILITY OF COT

We can also consider a slight variation of the potential, called the stability of a CoT. Given a prompt $\boldsymbol{x}$, CoT reasoning and answer pair $(\boldsymbol{c}, y)$ we define the stability of a sub-chain $\boldsymbol{c}_{<t}$ as

$$\text{stable}_N(\boldsymbol{c}_{<t}; \boldsymbol{x}, y) := \frac{1}{N} \sum_{n=1}^{N} \mathbb{1}_{\{y^{(n)}=y\}} \quad \text{where} \quad \left(y^{(n)}, \boldsymbol{c}_{\geq t}^{(n)}\right) \sim \text{LM}_\theta(\bullet|\boldsymbol{c}_{<t}, \boldsymbol{x})$$

with the slight variation that instead of considering the ground truth $y^*$, we now consider the reached final answer of the chain $\boldsymbol{c}$ as the target. I.e. the potential is a special of stability, when $y = y^*$. Stability measures how *determined* the final answer is throughout the reasoning process of the model. Somewhat surprisingly, we observe that correct answers do not necessarily always display higher stability, indicating that models can become convinced very early on in their reasoning about wrong answers. We display various stability curves in Fig. 16.

## A.5   COMPUTATIONAL COMPLEXITY OF POTENTIAL CALCULATION

The computational load to obtain the potential curve of a given prompt scales with the number of evaluations $N$ produced to estimate the potential and the number of points $N_{\text{chunks}}$ we calculate the potential for. More precisely, say for a generation from scratch we produce $T$ tokens, the total amount of tokens produced to estimate the potential curve will be given by

$$T_{\text{tot}} = N \sum_{i=1}^{N_{\text{chunks}}} \frac{i}{N_{\text{chunks}}} T \approx \frac{N N_{\text{chunks}} T}{2}$$

$N_{\text{chunks}}$ hence also gives us a way to control the amount of compute and we set it to moderate values of $N_{\text{chunks}} = 15$ for the quantitative evaluations, while we use higher values of around $N_{\text{chunks}} = 50$ for most figures, to obtain a more fine-grained understanding. Advanced inference techniques such as prefix-caching and continuous batching in vLLM further help to speed up the potential calculation, as all of the prefill is shared among the prompts, and every prompt can in theory be generated in parallel, thus allowing for continuous batching. This is different for the optimal CoT, where inference has to be done sequentially as things depend on eachother, making it significantly slower.

## A.6   CALCULATION OF OPTIMAL COT

We detail the recipe to calculate the optimal CoT in Algorithm 1 below:

---

**Algorithm 1** Generating potential-optimized CoTs

---

1: Initialize the CoT $c_{<t} \leftarrow \emptyset$
2: **while** the chosen candidate does not contain the answer **do**
3:     Sample $M$ candidate CoT chunks $c_{t:(t+T)}^{(m)} \overset{i.i.d.}{\sim} \text{LM}_\theta(\cdot \mid c_{<t}, x)$ of length $C$, for $m = 1, \ldots, M$
4:     Compute potentials $p_m \leftarrow \text{pot}_N(c_{<t+T}^{(m)}; x)$
5:     Select $\tilde{m} \leftarrow \arg\max_m p_m$
6:     Update $c_{<t+T} \leftarrow [c_{<t}, c_{t<(t+T)}^{(\tilde{m})}]$
7: **end while**

---

The computational complexity of the optimal CoT calculation is quite high with $\approx \frac{MNN_{\text{chunks}}T}{2}$ tokens needed, while also not enjoying the same benefits from prefix caching and continuous batching, as the potential curve calculation did. We thus do not recommend its usage in practical setting but view it more as a proof of concept.

A consequence of our greedy strategy is the following: The optimal CoT could degenerate to a "lucky guessing" answer when $M$ goes to infinity, given that the model has non-zero support across all tokens, as this would effectively minimize the objective. Since we are restricted to using a finite $M$ however in practice, we are implicitly regularizing the objective to sequences of sufficient probability mass, naturally circumventing this degeneracy. To make this regularization more explicit, one could potentially enhance the objective with a regularization term to require a given chain to meet a certain probability budget, eliminating this possibility. We observe however that $M$ being finite is enough to avoid such degenerate cases.
We do however want to stress that the optimal CoT does depend on $M$.

### A.7 MORE DETAILS ON SUMMARY STATISTICS

Here we provide the definitions for the statistics we used in Sec. 4. In all experiments we divide the CoT into 20 chunks, getting thus potential curves consisting of 20 points.

- **Insight:** We say that a given potential contains an insight if the difference between two consecutive chunks of CoT exceeds $40\%$, i.e. if one step of CoT raised the potential by at least $40\%$. We exclude the last two steps to make sure we don't count the late reasoning spikes as insights.

- **Tangent:** We define a potential to exhibit a tangent if the potential drops by at least $30\%$, not necessarily consecutively.

- **Guess:** We define late reasoning spikes or guesses as the case when the potential at the second to last step is smaller than $5\%$.

- **Monotonicity:** We call a potential monotone if its consecutive steps do not decrease by more than $10\%$.

