# OpenReview forum: "The Potential of CoT for Reasoning: A Closer Look at Trace Dynamics"
_ICLR.cc/2026/Conference — ICLR 2026 Poster_

### Official Review · Reviewer_m7iu · 2025-10-17

**Soundness:** 3
**Presentation:** 2
**Contribution:** 3
**Rating:** 6
**Confidence:** 3

**Summary:**

This paper introduces a novel metric called "potential" to analyze the reasoning process in large language models using Chain-of-Thought (CoT) prompting. By examining how the potential evolves over the course of a CoT on competition-level math problems, the authors identify several distinct patterns: "reasoning insights" where the potential sharply increases, "reasoning tangents" where it drops, and "lucky guesses" where the potential remains low until the very end .

**Strengths:**

(1) Novel and Intuitive Methodology: The core contribution, the "potential" metric (Eq. 1), offers a principled and intuitive method for quantifying progress within a CoT. This provides a valuable tool for moving beyond simple final-answer accuracy to analyze the intermediate reasoning steps.

(2) Strong Qualitative Analysis: The paper excels in its qualitative analysis, providing clear and well-annotated examples that connect the behavior of the potential curve to specific segments of the model's generated text.

(3) Interesting Empirical Findings: The study uncovers compelling and non-obvious dynamics in CoT reasoning.

**Weaknesses:**

(1) Estimating the potential metric requires sampling a large number of completions ($N=128$) at every intermediate CoT step to obtain a stable estimate. The manuscript does not address the resulting computational overhead, which may pose a serious limitation for scalability and broader adoption.

(2) The quantitative analysis in Table 1 depends on fixed, manually chosen thresholds to categorize behaviors such as “insights” (potential increase > 40%), “tangents” (potential drop > 30%), and “guesses” (potential < 5% at the penultimate step). The absence of justification for these cutoffs makes the reported statistics appear somewhat subjective.

(3) The empirical study centers almost entirely on competition-level mathematics problems (AIME) and on the Qwen family of models. It remains unclear whether the observed potential trajectories and reasoning patterns would extend to other tasks—such as commonsense reasoning or code generation—or to other model architectures.

(4) Algorithm 1 employs a brute-force approach that randomly samples and evaluates candidate CoT segments to maximize potential. This process is computationally inefficient and appears intended as a proof of concept. However, the manuscript does not discuss its scalability or the resources required to produce the results in Figure 3.

**Questions:**

Please refer to weaknesses.

---

> ### Author Response · Authors · 2025-11-24
> **Response**
>
> We thank the reviewer for taking the time to review our paper. We appreciate the feedback and have updated our paper to clarify and address the concerns. We will also answer the raised points one by one in the following:
>
> * *Estimating the potential metric requires sampling a large number of completions (N=128) at every intermediate CoT step to obtain a stable estimate. The manuscript does not address the resulting computational overhead, which may pose a serious limitation for scalability and broader adoption.*
>
>     We thank the reviewer for bringing up this point and agree that estimating the potential can be expensive, but want to also highlight some aspects that reduce the cost quite substantially.
>     (1) In this paper we have used a high number $N=128$ to really ensure that we have very accurate potentials to facilitate the detailed analysis performed in the work. In practice however we do observe that smaller numbers such as $N=16$ suffice to obtain a good picture of the potential. (2) The number of chunks that we divide the CoT into controls the granularity of the potential, the higher the number of chunks the higher the amount of compute needed (we detail the complexity in Appendix A.5). For some applications, a small number of chunks might suffice, e.g. for instance for partial reward computation, it might be sufficient to divide the CoT into 2 or 3 sections to perform better credit assignment already. (3) Moreover, the potential computation for a given CoT can be performed completely in parallel across all chunks, with prefix-caching alleviating some of the burden from repetitive computations. Indeed we leverage this fact by using vLLM with continuous batching and prefix-caching activated [1]. While of course still not as cheap as a single generation, we believe that the incurred computational cost does not posit a blocker for most users. We have added a paragraph in Appendix A.5 that explains the computational costs described here.
> * *The quantitative analysis in Table 1 depends on fixed, manually chosen thresholds to categorize behaviors such as “insights” (potential increase > 40%), “tangents” (potential drop > 30%), and “guesses” (potential < 5% at the penultimate step). The absence of justification for these cutoffs makes the reported statistics appear somewhat subjective.*
>
>     We agree with the reviewer that setting such parameters in estimating the properties of the potential curve is non-trivial and most likely never fully principled. We have chosen those values informed by the qualitative analysis we performed on several examples, where we have identified the different behaviors. We acknowledge however that these values are not universal and recommend users to examine some examples of their use-case to perform the quantitative analysis, similar to how we have done it in our work.
> * *The empirical study centers almost entirely on competition-level mathematics problems (AIME) and on the Qwen family of models. It remains unclear whether the observed potential trajectories and reasoning patterns would extend to other tasks—such as commonsense reasoning or code generation—or to other model architectures.*
>
>     We agree with the reviewer and to further strengthen the results, we have performed more experiments to analyze the potential by adding models Llama-3.1-8B and 70B, as well as datasets MATH-500 (see Table 2, Figure 12 and 13) with both qualitative and quantitative results. We point the reviewer to the response to all reviewers for a brief discussion of the results.
> * *Algorithm 1 employs a brute-force approach that randomly samples and evaluates candidate CoT segments to maximize potential. This process is computationally inefficient and appears intended as a proof of concept. However, the manuscript does not discuss its scalability or the resources required to produce the results in Figure 3.*
>
>     We thank the reviewer for raising this point and have added a section detailing the cost. As the reviewer suspects, computing the optimal CoT is very costly, as in contrast to the potential calculation, this is an inherently sequential task, more strongly limiting the usage of continuous batching and prefix-caching as it essentially only works “within the chunk” now as opposed to “across all chunks”. Indeed we view the optimal CoT experiments more as a proof of concept and would not recommend users to rely on scaling it too strongly.
>
>
> [1] https://docs.vllm.ai/en/latest/features/automatic_prefix_caching/

---

### Official Review · Reviewer_v32n · 2025-10-30

**Soundness:** 3
**Presentation:** 3
**Contribution:** 3
**Rating:** 6
**Confidence:** 3

**Summary:**

The paper studies chain-of-thought (CoT) reasoning in large language models using a new measure called “potential,” which quantifies how each reasoning step contributes to getting the correct answer. The authors analyze competition-level math problems to reveal how CoT progress fluctuates — sometimes increasing sharply with key insights, other times dropping due to reasoning tangents or random guesses. They further explore how partial CoT from stronger models can boost weaker ones, suggesting reasoning transfer across architectures.

**Strengths:**

The paper addresses an important gap in understanding how CoT actually helps models reason, not just appear to. Introducing “potential” as a metric is novel and provides a quantitative way to evaluate reasoning progress. The experiments on AIME problems are well-motivated and the qualitative analysis of “reasoning tangents” and “insights” is compelling. The study also goes beyond introspection by testing CoT transferability, showing interesting empirical results that small parts of reasoning from stronger models can unlock weaker ones

**Weaknesses:**

The proposed concept of potential is not rigorously justified beyond empirical correlation, and its interpretation remains vague. The assumption that higher potential implies genuine reasoning progress is too strong, as sampling-based estimation might simply capture distributional artifacts. The analysis often reads descriptive rather than explanatory; the paper reports patterns (insights, jumps, tangents) without providing mechanisms or theoretical grounding. The heavy reliance on visual examples and qualitative claims weakens the argument, since the method’s reliability is unclear. The “transferability” results are interesting but confounded — providing partial CoT is effectively giving a hint, so improved performance does not prove reasoning generalization. There is also limited diversity in tasks and model families; focusing only on AIME math and Qwen variants narrows generality. The paper overstates human-analogy comparisons despite limited evidence. Overall, while the work is original and thought-provoking, it lacks methodological rigor and over-interprets its empirical patterns.

**Questions:**

see above

---

> ### Author Response · Authors · 2025-11-24
> **Response**
>
> We thank the reviewer for taking the time to review our manuscript. We have updated the text to incorporate the provided feedback but will also answer each point one by one in the following:
>
>
> * *The proposed concept of potential is not rigorously justified beyond empirical correlation, and its interpretation remains vague. The assumption that higher potential implies genuine reasoning progress is too strong, as sampling-based estimation might simply capture distributional artifacts.*
>
>     We thank the reviewer for this comment as it suggests that there was some misunderstanding, we do not want to claim that a higher potential implies genuine reasoning. The potential allows us to characterize what steps in the CoT lead to an increase or decrease in the probability of success. We then analyze several trajectories in more detail and observe that increases in the potential sometimes indeed correspond to key insights, while in other cases it is unclear why the particular step was essential to the model. We thus do not posit that a higher potential always implies reasoning, but rather allows us to pin-point which section of CoT were important to the model. If any part of the text suggested a stronger interpretation, we are happy to amend it.
> * *The analysis often reads descriptive rather than explanatory; the paper reports patterns (insights, jumps, tangents) without providing mechanisms or theoretical grounding. The heavy reliance on visual examples and qualitative claims weakens the argument, since the method’s reliability is unclear.*
>
>     To some degree we agree with the reviewer; we use the potential to characterize the parts of chain-of-thought that influence the probability of success the most and thereby observe specific patterns exhibited by these potential curves. These patterns deviate from what should be expected in theory, as we show in Proposition 1, which instead would predict a monotonic potential. We agree that we do not provide theoretical arguments as to why such deviations occur, as this is quite a difficult task. To still provide some insights, we investigate in strong detail how such deviations look and perform a qualitative analysis of various reasoning chains. While these results are only qualitative, we still believe that they provide a better understanding as to why such deviations occur and in our view do not weaken our arguments.
>
> * *The “transferability” results are interesting but confounded — providing partial CoT is effectively giving a hint, so improved performance does not prove reasoning generalization.*
>
>     We believe that this is precisely the core point of our experiment; we want to understand if the smaller model can leverage the insights (or hints) of the stronger model to also achieve better generalization. We want to especially highlight that such a “transfer” already occurs at very small amounts of 10-20% of partial CoT.  It is not obvious that smaller models are able to leverage such hints or insights as it could potentially be beyond their capacity. This experiment thus shows that such insights indeed transfer between models.
> * *There is also limited diversity in tasks and model families; focusing only on AIME math and Qwen variants narrows generality. The paper overstates human-analogy comparisons despite limited evidence.*
>
>     We agree with the reviewer and to further strengthen the results, we have performed more experiments to analyze the potential by adding models Llama-3.1-8B and 70B, as well as datasets MATH-500 (see Table 2, Figure 12 and 13) with both qualitative and quantitative results. We point the reviewer to the response to all reviewers for a brief discussion of the results.

---

### Official Review · Reviewer_1SnR · 2025-11-01

**Soundness:** 3
**Presentation:** 3
**Contribution:** 2
**Rating:** 4
**Confidence:** 4

**Summary:**

The authors endeavor the investigate which parts of the CoT are most "important" in the model's reasoning and eventual production of the answer. To do so, the authors define the notion of potential, which, given some partial CoT, is defined as the probability that the model will output the correct answer after generating the completion of the CoT. Thus, the authors can ask which parts of the CoT have the largest potential or the largest change in potential.

**Strengths:**

- The authors characterize a new way to study which parts of the CoT are most impactful/useful in the model's reasoning via a new metric they call "potential".
 - Some interesting observations are made that connect the changes in the potential with different parts/strategies in the CoT.
 - The paper is largely well-written.

**Weaknesses:**

- The implications and future applications of the potential metric are unclear (or they seem rather limited). Additional quantitative analysis may enable us to draw more general observations about the reasoning of LLMs. The conclusion is missing a discussion on future work, which could have alleviated this concern.
 - Experiments were focused on math competition examples, and it's unclear if similar observations would be drawn in other domains.

**Questions:**

The notion of potential is very closely related to the value function in actor-critic models in RL, but the paper doesn't really draw this connection. A vague connection is made at the end of Section 2, but the similarity between the two notions warrants further discussion and elaboration.

Algorithm 1 describes a "greedy" strategy for finding the optimal CoT. However, there may exist more "globally optimal" CoTs that find the correct answer faster than the CoT produced by Algorithm 1, while at the same time, do not increase the potential as quickly. But I suppose the probability of the CoT is an important consideration, as there exists the possibility of a "degenerate" CoT where the model guesses the correct answer immediately, which would result in an immediate jump in the potential at the very beginning of the CoT (due to lucky guessing). As such, the result of Algorithm 1 is sensitive to the choice of M, where in the limit as M goes to infinity, the algorithm would eventually recover this degenerate (lucky guess) CoT.

Some of the key observations were made in the qualitative analysis, such as the one in Figure 6, where the portion of the CoT that is more difficult for humans actually corresponds to a small increase in potential, whereas other portions that seem quite simple to humans are associated with larger jumps in potential. It is unclear to what extent these observations generalize to other examples without any accompanying quantitative evidence. How often does the behavior observed in Figures 7 and 8 occur?


More detailed comments and questions follow. While the paper is largely well-written, there are a small number of grammatical and stylistic errors. I include some of them from the first three sections in the list below. This list is not comprehensive, so I encourage the authors to carefully read through the paper and correct all such errors.

Line 28: A weaker what? Is this a typo?

Line 31: Grammatical error: there should be a complete clause following "that", such as "...highlighting that the grass is green." Here, note that "grass is green" can stand on its own as a complete sentence. However, "a large part of the mechanics underpinning reasoning transfer" is not a complete sentence/clause.

Line 36: "lead to" -> "led to"

There are a number of claims in the introduction that are stated without supporting citations. For example "[CoT] reasoning (Wei et al. 2023) has [led] to several breakthroughs in domains spanning mathematics"; and similarly for "...coding." There are several others.

Line 298: "Model size also seems to surpress [guessing behaviour] more, which is expected since larger models generally tend to perform better"  Wouldn't this only be expected if the model's final answers are faithful with respect to their CoT? For example, it is possible that larger models are better at finding the correct answer *latently* (in a fashion that is not verbalized in their CoT), in which case, they may be able to more accurately find the correct answer via the guessing tactic.

Line 306: Why is the pass@k metric prone to suffer from guessing behavior?

---

> ### Author Response · Authors · 2025-11-24
> **Response 1/2**
>
> We thank the reviewer for taking the time to review our paper. We found the comments and questions very helpful to improve the quality of our work and have subsequently updated our manuscript . In the following, we will also address the reviewer’s concerns one by one.
>
>
> * *The implications and future applications of the potential metric are unclear (or they seem rather limited). Additional quantitative analysis may enable us to draw more general observations about the reasoning of LLMs. The conclusion is missing a discussion on future work, which could have alleviated this concern.*
>
>     We agree with the reviewer that the previous manuscript was unclear regarding future applications of the potential metric but we will happily push back and provide several avenues that we envision our work to be useful:
>     * More generally, we view the potential metric as a tool to gain a better understanding of chain-of-thought traces, which have become extremely long in recent months due to advances in reinforcement learning. Pinning down which sections of CoT contributed positively (or negatively) in an automatic fashion can thus be very helpful to understand with what current models struggle. We believe that this will apply to all future models, since the potential metric is a very general tool.
>     * The potential can be used as a partial reward for a chunk of CoT, i.e. we can perform more fine-grained credit assignment in the setting of verifiable final rewards such as mathematics or coding. Incorporating the potential in reinforcement learning thus makes for exciting future work and some concurrent papers have already taken a similar route, e.g. [1, 2].
>     * We believe that our results on the transferability of CoT between models makes for interesting future work, where partial CoTs from strong models can be used to address the sparse reward problem in RL training. Providing the model with partial CoT for problems it cannot solve could potentially unblock learning and at least allow for some training signal.
> * *Experiments were focused on math competition examples, and it's unclear if similar observations would be drawn in other domains.*
>
>     We agree with the reviewer and to further strengthen the results, we have performed more experiments to analyze the potential by adding models Llama-3.1-8B and 70B, as well as datasets MATH-500 (see Table 2, Figure 12 and 13) with both qualitative and quantitative results. We point the reviewer to the response to all reviewers for a brief discussion of the results.
> * *The notion of potential is very closely related to the value function in actor-critic models in RL, but the paper doesn't really draw this connection. A vague connection is made at the end of Section 2, but the similarity between the two notions warrants further discussion and elaboration.*
>
>    We thank the reviewer for pointing out this connection and we have updated the text to reflect this. Indeed there is quite a strong connection,  the value function aims to estimate the future reward given a state $ s_t $, whereas the potential essentially estimates the average reward directly in a Monte-Carlo fashion.
> * *Algorithm 1 describes a "greedy" strategy for finding the optimal CoT. However, there may exist more "globally optimal" CoTs that find the correct answer faster than the CoT produced by Algorithm 1, while at the same time, do not increase the potential as quickly. But I suppose the probability of the CoT is an important consideration, as there exists the possibility of a "degenerate" CoT where the model guesses the correct answer immediately, which would result in an immediate jump in the potential at the very beginning of the CoT (due to lucky guessing). As such, the result of Algorithm 1 is sensitive to the choice of M, where in the limit as M goes to infinity, the algorithm would eventually recover this degenerate (lucky guess) CoT.*
>
>     This is an interesting insight of the reviewer and we agree that theoretically, the optimal CoT would degenerate to a "lucky guessing" answer when M goes to infinity, given that the model has non-zero support across all tokens. Since we are restricted to using a finite M, we are implicitly regularizing the objective to sequences of sufficient probability mass, naturally circumventing this degeneracy. To make this regularization more explicit, one can enhance the objective with a regularization term to require a given chain to meet a certain probability budget. Given the finiteness of our experiments we don’t view this as necessary in our experiments but for theoretical considerations we completely agree.
>
>
> 1. https://arxiv.org/pdf/2505.23564
> 2. https://arxiv.org/pdf/2506.11902

---

> > ### Author Response · Authors · 2025-11-24
> > **Response 2/2**
> >
> > * *Some of the key observations were made in the qualitative analysis, such as the one in Figure 6, where the portion of the CoT that is more difficult for humans actually corresponds to a small increase in potential, whereas other portions that seem quite simple to humans are associated with larger jumps in potential. It is unclear to what extent these observations generalize to other examples without any accompanying quantitative evidence. How often does the behavior observed in Figures 7 and 8 occur?*
> >
> >     This is an interesting observation by the reviewer that the amount of increase in the potential varies according to human-perceived difficulty, i.e. the more difficult insights seem to lead to less increase, compared to the seemingly easier steps. We want to clarify however that we have not made this claim in the text, rather the take-away from our results is that (1) such sudden increases happen frequently (see Table 1 and 2), regardless of whether it is difficult or not from a human perspective and (2) these increases can both be interpretable (i.e. correspond to a key insight) or more model-specific (i.e. it is not obvious why the step leads to a large increase in potential). Beyond this, we do not want to make more claims as they indeed are not substantiated by empirical evidence. If the text misled the reviewer in any way, we will happily amend the corresponding sections.
> > * *More detailed comments and questions follow. While the paper is largely well-written, there are a small number of grammatical and stylistic errors. I include some of them from the first three sections in the list below. This list is not comprehensive, so I encourage the authors to carefully read through the paper and correct all such errors.*
> >
> >     We thank the reviewer for taking the time to note down some of the mistakes. We have gone over the text again and improved its quality.
> > * *There are a number of claims in the introduction that are stated without supporting citations. For example "[CoT] reasoning (Wei et al. 2023) has [led] to several breakthroughs in domains spanning mathematics"; and similarly for "...coding." There are several others.*
> >
> >     We thank the reviewer for pointing this out, we have extended our bibliography to support our statements more, especially in the introduction.
> > * *Line 298: "Model size also seems to surpress [guessing behaviour] more, which is expected since larger models generally tend to perform better" Wouldn't this only be expected if the model's final answers are faithful with respect to their CoT? For example, it is possible that larger models are better at finding the correct answer latently (in a fashion that is not verbalized in their CoT), in which case, they may be able to more accurately find the correct answer via the guessing tactic.*
> >
> >     This is an interesting point! We agree that the larger models could perform more computations latently, without verbalizing the path to the answer, making it seem like the model is guessing the answer in the end. The potential metric would however reflect this behavior, as the probability of success would still go up throughout the chain-of-thought, even if the tokens carry no semantic meaning. The probability of success would hence not spike in the final part of the CoT in this case, as the preceding tokens do meaningful computations.
> > * *Line 306: Why is the pass@k metric prone to suffer from guessing behavior?*
> >
> >     Upon re-reading, we realize that the phrasing was misleading. Not pass@k itself as a general metric is susceptible to being gamed by guessing but rather pass@k in combination with final-answer verification only. With mathematics being the focus of our work, this can indeed be the case as AIME answers are standardized to be between 0-1000. No other parts of the response except for the final answer are verified, allowing the model to thus succeed if the final answer is guessed correctly. This is in contrast with other areas, such as for instance coding, where a solution is usually verified through a number of unit tests. With a high number of unit tests, writing a program that passes all of them becomes increasingly unlikely and pass@k in this case is not prone to guessing behavior.

---

> > > ### Comment · Reviewer_1SnR · 2025-11-26
> > >
> > > I thank the authors for their thoughtful responses to the questions raised in the review. I also appreciate the addition of future work in the Conclusion. I recommend that the authors split this discussion into a second paragraph in this section and perhaps add further details (why not include the concurrent work cited in the response?).
> > >
> > > I appreciate the additional experiments on Llama models and on the MATH-500 dataset. However, my initial concern was whether the findings generalize to domains aside from mathematics (such as coding, non-mathematical reasoning, etc). It does not seem like the idea of the potential is inherently restricted to the domain of mathematical problem solving.
> > >
> > > Given that the algorithm for generating potential-optimized CoTs is dependent on M, I would recommend the authors explicitly state the possibility discussed in the review and modify Algorithm 1 to have M as an explicit parameter.
> > >
> > > Given the revisions so far, I will raise the contribution score in the review.

---

> > > > ### Author Response · Authors · 2025-12-02
> > > > **Response**
> > > >
> > > > We thank the reviewer for the prompt response and engaging with our rebuttal!
> > > >
> > > > * *I recommend that the authors split this discussion into a second paragraph in this section and perhaps add further details (why not include the concurrent work cited in the response?).*
> > > >
> > > >     We agree with the reviewer and have split the discussion section by adding a paragraph solely dedicated to future work. We added the provided citations in the reply to the main text, we apologize for not doing so before.
> > > > * *I appreciate the additional experiments on Llama models and on the MATH-500 dataset. However, my initial concern was whether the findings generalize to domains aside from mathematics (such as coding, non-mathematical reasoning, etc). It does not seem like the idea of the potential is inherently restricted to the domain of mathematical problem solving.*
> > > >
> > > >     We understand the reviewer’s concern now better and as a consequence we performed more experiments on HumanEval [1] (coding benchmark) and GPQA-Diamond [2] (more common reasoning). Due to space reasons we had to move all results to the Appendix. We also had to limit the evaluation to a smaller number of models (Qwen2.5 and Llama-3.1 mostly) due to time constraints. We show quantitative results in Table 3 and 4 as well as additional qualitative results in Figure 14 and 15. Interestingly, we find that on average, the models seem more stable on coding benchmarks, with their CoT displaying more monotonic behaviour and in general less tangents. Still, we observe both reasoning insights and tangents also for coding tasks as shown in Figure 14 and 15. For GPQA we find trends more consistent with AIME, most likely because more questions are very difficult compared to HumanEval. As expected due to GPQA being a multiple choice benchmark with only four choices, we observe a significantly higher guessing rate.
> > > >
> > > > * *Given that the algorithm for generating potential-optimized CoTs is dependent on M, I would recommend the authors explicitly state the possibility discussed in the review and modify Algorithm 1 to have M as an explicit parameter.*
> > > >     We have adapted the text to really stress that the optimized CoT does depend on M, thank you for pointing this out. We have incorporated the discussion from the rebuttal into the paper, see Appendix A.6 for the text.
> > > >
> > > >
> > > >
> > > > While we’re aware that the reviewer cannot reply or make any changes anymore, we still hope that the changes to our paper and the new experiments convince the reviewer that the potential is useful across benchmarks.
> > > >
> > > > [1] https://arxiv.org/abs/2107.03374
> > > >
> > > > [2] https://arxiv.org/abs/2311.12022

---

### Author Response · Authors · 2025-11-24
**To all reviewers**

We thank all the reviewers for taking the time to carefully review our manuscript. A common concern shared among reviewers was our method not being evaluated on enough models and datasets. We have now performed more experiments on the Llama-3.1 family of models, and have further added both quantitative and qualitative results on MATH-500 (Table 2, Figure 12 and 13 in the current version of the paper). We find that the same trends observed on AIME-2024 and 2025 also hold on MATH-500, with the Llama family displaying more erratic potentials on average, most likely due to its inferior performance in general on these benchmarks. We further identify the presence of reasoning tangents and insights for the Llama model family too, also in case of MATH-500, we display two examples in Figure 12 and 13 in the Appendix.

---

### Author Response · Authors · 2025-12-04
**Summary for area chair**

Dear Area Chair,

We understand that you are taking over under difficult circumstances. To facilitate your assessment, we summarize the updates made in the rebuttal below:


* **Additional experiments:** As requested by some reviewers, we have significantly expanded the range of benchmarks considered, by adding MATH-500, HumanEval (coding) and GPQA-Diamond (STEM-reasoning) to further showcase that our technique is **applicable across many types of domains**. We have also added results on the Llama-3.1 family to further underline the **generality of our results across model families**. We point to Table 2, 3 and 4 for quantitative, as well as to Figure 12, 13, 14 and 15 for qualitative results on these new benchmarks. The results largely confirm the trends we observed for AIME-2024 and AIME-2025.
* **Future applications:** One reviewer inquired about future applications of our technique, which was not discussed in detail in the first version of our paper. We have updated it now, outlining the following points:
    * The potential metric can lead to a better understanding of chain-of-thought traces, allowing one to pin down which sections of CoT contributed positively (or negatively) in an automatic fashion.
    * The potential can be used as a partial reward for a chunk of CoT, i.e., we can perform more fine-grained credit assignment in the setting of verifiable final rewards. Incorporating the potential in reinforcement learning thus makes for exciting future work and some concurrent papers have already taken a similar route, e.g. [1, 2].
    * We believe that our results on the transferability of CoT between models makes for interesting future work. For instance, during RL training, one can provide the model with partial CoT from a stronger model (which can result in high potential for the weaker model as we showed in Fig. 8), which could unblock learning for difficult problems and allow for some training signal. Indeed concurrent work [3] has obtained some preliminary positive results.
* **Clarifications:** We have updated the paper to clarify several aspects such as the computational complexity of the potential (Appendix A.5), the dependency of the optimized CoT on the number of samples drawn (Appendix A.6) and fixed several citation and grammatical issues.



Reviewer 1SnR engaged with our rebuttal, raising the contribution score to 3. Reviewer 1SnR clarified that additional experiments on non-mathematical tasks would satisfy them (up to this stage we only performed more experiments on MATH-500), which prompted us to add more experiments on HumanEval (coding) and GPQA-Diamond (STEM). Unfortunately, Reviewer 1SnR could not reply anymore to our new experiments, due to the change in the reviewing system. We hope that the AC can take this into account in the final decision.



[1]https://arxiv.org/abs/2505.23564

[2] https://arxiv.org/abs/2506.11902

[3] https://arxiv.org/abs/2506.18110

---

### Meta-Review · Area_Chair_2ocy · 2026-01-07

**Summary:**

The paper proposes a "potential" metric to quantify the contribution of individual Chain-of-Thought (CoT) steps, identifying patterns like reasoning spikes, tangents, and lucky guesses. Reviewers found the metric novel and the qualitative analysis insightful. The primary concerns were the narrow experimental scope (initially limited to competition math and Qwen models), the high computational cost of the sampling-based metric, and the subjective nature of the thresholds used to classify reasoning patterns. The authors significantly expanded their experiments during the rebuttal to address the scope issues.

**Reviewer Concerns:**

Reviewer Concerns
Addressed:
- Generalizability: The most significant concern (shared by all reviewers) was the reliance on a single domain. The authors added experiments on HumanEval (coding), GPQA, and MATH-500, along with Llama-3.1 models, effectively demonstrating transferability beyond the initial setup.
- Clarifications: The connection between "potential" and RL value functions was clarified, and a discussion on future applications was added to the conclusion, satisfying Reviewer 1SnR.

Outstanding:
- Computational Cost: While the authors discussed mitigation strategies (e.g., prefix caching), the method inherently requires heavy sampling (N=128), which remains a barrier to scalability (Reviewer m7iu).
- Subjectivity: The thresholds defining "insights" vs. "tangents" remain manually selected based on qualitative observation rather than rigorous theoretical derivation (Reviewer v32n, m7iu).

**Reviewer Scores:**

Reviewer 1SnR: 4 -> 6
Reviewer v32n: 6 -> 6
Reviewer m7iu: 6 -> 6

---

### Decision · Program_Chairs · 2026-01-26

Accept (Poster)